# Temporal Guidance for Large Language Models

## Abstract

Contrastive Decoding (CD) enhances the generation quality of large language models (LLMs) but incurs significant additional computational overhead due to the need for an auxiliary model. Existing internal self-contrastive decoding methods, such as Decoding by Contrasting Layers (DoLa), focus on discrepancies across different layers, which are notably unstable on small-scale models. In this work, based on the observation that LLMs exhibit local preferences, we propose a novel contrastive guidance strategy along the temporal dimension, namely **Temporal Guidance (TeGu)**. Our method ingeniously leverages Multi-Token Prediction (MTP) to construct weaker amateur predictions for model self-contrast. To standardize the implementation of this mechanism, we further introduce a lightweight Conditional MTP Projector (cMTPP), which avoids maintaining multiple independent networks as required by other MTP modules. Across various model series and benchmarks, TeGu achieves significant performance improvements while maintaining low additional memory consumption and computational overhead. Here is our anonymous code.

## 1. Introduction

Large Language Models (LLMs) (Vaswani et al., 2017; Radford et al., 2018; Touvron et al., 2023; Yang et al., 2025a; Guo et al., 2025) have demonstrated remarkable capabilities across a wide range of tasks, from complex reasoning to creative writing. However, the standard autoregressive generation process, which typically relies on greedy or nucleus sampling (Holtzman et al., 2019b), often suffers from inherent limitations such as repetition loops, generic responses, and hallucinations (Holtzman et al., 2019b; Welleck et al., 2019; Ji et al., 2023; Lin et al., 2022). To address these

issues, Contrastive Decoding (CD) (Li et al., 2023; O'Brien & Lewis, 2023) strategies have emerged as a promising paradigm. By contrasting the logits of a capable "expert" model with those of a less capable "amateur" model, CD effectively amplifies signals associated with reasoning and factuality while suppressing common language priors.

Standard CD requires the simultaneous use of two independent models, one large and one small, and necessitates maintaining two sets of KV cache. This is highly cumbersome and significantly increases VARM usage and inference latency. To mitigate this, internal contrastive approaches (Chuang et al., 2023; Yu et al., 2025b; Zhu et al., 2025), like Decoding by Contrasting Layers (DoLa) (Chuang et al., 2023) attempt to derive the amateur signal from the model's own shallow layers. However, DoLa suffers from severe instability on smaller-scale architectures (Chuang et al., 2023), often leading to a performance collapse in logic-intensive tasks like coding. This suggests that the layer-wise disentanglement assumption of DoLa may not hold universally across different model scales.

In this paper, we propose **Temporal Guidance (TeGu)**, a robust and efficient decoding strategy that exploits the temporal dimension of LLMs. Our method is grounded in the observation that LLMs exhibit a strong locality bias, relying heavily on immediate tokens for precise generation (Liu et al., 2024; Khandelwal et al., 2018; Xiao et al., 2023; Press et al., 2021). In an MTP setting (Gloeckle et al., 2024; Samragh et al., 2025; Cai et al., 2024), auxiliary heads are tasked with predicting a target token $x_t$ based on a historical context $x_{<t-k}$ (where $k$ is the temporal offset). Deprived of the critical immediate context, these heads naturally produce a high-entropy, context-deficient distribution that perfectly fits the definition of an "amateur."

To generalize TeGu to mainstream LLMs that lack native MTP support, we introduce a minimalist and efficient Conditional MTP Projector (cMTPP). Unlike standard MTP implementations that rely on several independent heads (Gloeckle et al., 2024; Samragh et al., 2025; Cai et al., 2024; Liu et al., 2025), our cMTPP incorporates the future offset as a conditional input to modulate historical hidden states, which are then projected into the vocabulary space using the frozen language model head. To prevent the distribution of the MTP from deviating from the original NTP (Next-Token

[1]Anonymous Institution, Anonymous City, Anonymous Region, Anonymous Country. Correspondence to: Anonymous Author <anon.email@domain.com>.

Preliminary work. Under review by the International Conference on Machine Learning (ICML). Do not distribute.

Prediction) distribution of the model, we train the projector using a combined objective of Cross-Entropy and Knowledge Distillation (Hinton et al., 2015; Kim & Rush, 2016; Gu et al., 2023; Agarwal et al., 2023). During the inference phase, TeGu guides the generation process by comparing the predictions of NTP with those of MTP. We validate TeGu across multiple model families and scales.

Our contributions are summarized as follows:

- We introduce **Temporal Guidance (TeGu)**. Moving beyond the model-level and layer-level contrasts utilized in prior works, TeGu establishes a new paradigm by exploiting the temporal dimension of LLMs. It leverages historical predictions from MTP modules as natural amateurs, eliminating the need for external models or manual layer selection.

- We design a minimalist Conditional MTP Projector (cMTPP) to generalize TeGu to LLMs lacking native MTP support. cMTPP is aligned with the frozen backbone via knowledge distillation, ensuring seamless integration and preventing distribution shift issues.

- Comprehensive experiments show that TeGu outperforms standard Greedy decoding, standard CD and DoLa across mathematics, coding, and instruction-following benchmarks. TeGu achieves superior inference efficiency compared to other CD approaches.

## 2. Related Work

### 2.1. Contrastive Decoding and Guidance Mechanisms

Contrastive Decoding (CD) has emerged as a powerful paradigm to enhance the generation quality of Large Language Models (LLMs) by amplifying the signal from an "expert" distribution while suppressing a flawed "amateur" distribution. Originally proposed by Li et al. (2023), CD utilizes a smaller model as the amateur to penalize generic and repetitive patterns. This framework has been extended to reasoning tasks (O'Brien & Lewis, 2023) and combined with Classifier-Free Guidance (CFG) (Ho & Salimans, 2022; Sanchez et al., 2023) to steer models towards desired prompts. MTI (Yang et al., 2025b) only intervenes on high-uncertainty tokens to reduce computational load. Recent iterations have further refined this through adaptive masking (Li et al., 2025a), uncertainty-aware mechanisms (Lee et al., 2025; Ding et al., 2024), and multi-amateur strategies (Sen et al., 2025). Other works integrate distillation into the decoding process (Phan et al., 2024) or utilize prompt-based contrasts (Lv et al., 2024) to sharpen model outputs. Su et al. (2022); Su & Collier (2022) proposed Contrastive Search to enforce isotropy in representation space, while Yang et al. (2024) introduced a lightweight "Frustratingly Simple Decoding" method to penalize repetition.

Despite its effectiveness, standard CD suffers from memory overhead due to dual-model requirements. To address this, internal contrastive methods derive amateur signals from the model itself. Chuang et al. (2023) introduced DoLa to contrast deep and shallow layers. DoLa, LACD (Yu et al., 2025b) and Delta-Contrastive Decoding (Huang & Chen, 2025) effectively mitigate hallucinations. Similar layer-wise contrastive strategies include LayerCake (Zhu et al., 2025), Adaptive Layer-Wise decoding (Zhou et al., 2025), and Active Layer-Contrastive Decoding (Zhang et al., 2025a). Others explore contrasts via pruned self-models (Yu et al., 2025a), retrieval heads (Gema et al., 2024), token-wise cross-layer entropy (Wu et al., 2025), or auto-contrastive mechanisms (Gera et al., 2023).

A major focus of decoding strategies is mitigating hallucinations and resolving knowledge conflicts. Techniques like CAD (Shi et al., 2024), CoCoA (Khandelwal et al., 2025), CAAD (Nguyen et al., 2025), and AdaCAD (Wang et al., 2025) adjust decoding based on context awareness. Others, such as SLED (Zhang et al., 2024), ICD (Zhang et al., 2025b), and HICD (Jiang et al., 2025), induce or evolve logits to penalize hallucinations, while SH2 (Kai et al., 2024) introduces hesitation for truthfulness.

### 2.2. Multi-Token Prediction

Multi-Token Prediction (MTP) extends standard autoregressive modeling by predicting multiple future tokens simultaneously (Gloeckle et al., 2024). This paradigm is adopted by advanced models like Qwen3 (Yang et al., 2025a), DeepSeek-R1 (Guo et al., 2025) and MiMo (Xiaomi et al., 2025; Xiao et al., 2026). Research has optimized MTP using register tokens (Gerontopoulos et al., 2025), future summaries (Mahajan et al., 2025), and independent decoding heads (Liu et al., 2025; Samragh et al., 2025), or by modeling token order (Zuhri et al., 2025).

The primary application of MTP heads has been Speculative Decoding (Leviathan et al., 2023b) to accelerate inference. Frameworks like Medusa (Cai et al., 2024) and EAGLE (Li et al., 2024; 2025b) use these heads to draft tokens for verification. FastMTP (Cai et al., 2025) aligns training with inference for further speedups.

## 3. Method

### 3.1. Preliminaries

**Autoregressive Generation.** Given a sequence of tokens $x_{<t} = \{x_1, \ldots, x_{t-1}\}$, a standard causal LLM models (Vaswani et al., 2017; Touvron et al., 2023; Yang et al., 2025a; Guo et al., 2025) the probability of the next token $x_t$ as $P_\theta(x_t|x_{<t})$. The generation process typically selects $x_t$ by sampling from this distribution or choosing the token with the maximum probability.

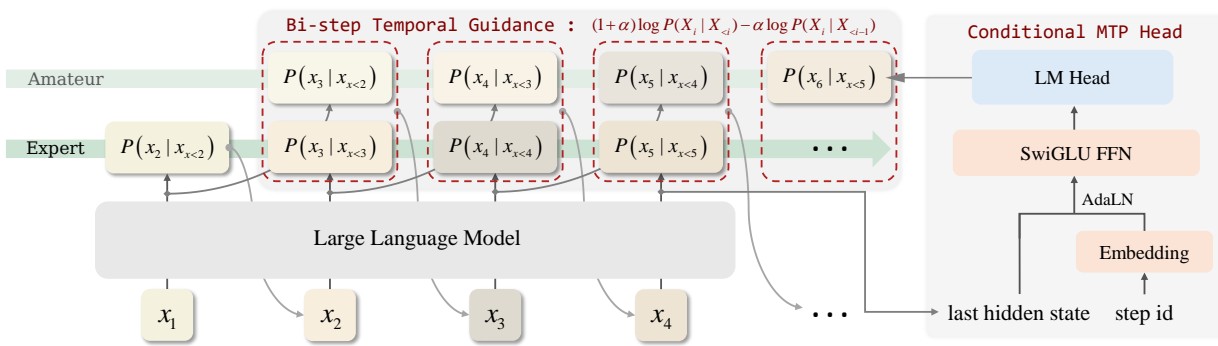

*Figure 1.* **Overview of Temporal Guidance (TeGu).** Left: TeGu leverages the contrast between the MTP outputs from previous steps (Amateur) and the NTP from the current step (Expert). It performs logits update using the formula $(1 + \alpha) \log P_{\exp} - \alpha \log P_{\mathrm{amt}}$. Subsequently, the next token is obtained based on the updated logits. Right: The Conditional MTP Projector employs AdaLN to inject the future time step offset (Step ID) into the hidden states, which are then processed through a SwiGLU FFN. The frozen base language model head is reused to obtain the MTP outputs.

**Multi-Token Prediction.** Multi-Token Prediction (MTP) (Gloeckle et al., 2024; Samragh et al., 2025; Cai et al., 2024; Liu et al., 2025) predicts a sequence of $n$ future tokens $\{x_t, \ldots, x_{t+n-1}\}$ simultaneously given context $x_{<t}$. Typically implemented by appending $n$ independent heads to a shared backbone, it aggregates losses from all heads: $\mathcal{L}_{\mathrm{MTP}} = \sum_{k=0}^{n-1} \mathcal{L}_{\mathrm{CE}}(P(x_{t+k}|x_{<t}), x_{t+k})$. This paradigm enhances long-range planning, and during inference is typically utilized for speculative decoding (Xia et al., 2023; Leviathan et al., 2023a; Cai et al., 2024; Li et al., 2024).

**Contrastive Decoding.** Contrastive Decoding (CD) (Li et al., 2023; O'Brien & Lewis, 2023) and Classifier-Free Guidance (CFG) (Ho & Salimans, 2022; Sanchez et al., 2023) amplify an expert distribution $P_{\exp}$ by contrasting it with an amateur $P_{\mathrm{amt}}$:

$$\log P_{\mathrm{guided}} = \log P_{\exp} + \alpha(\log P_{\exp} - \log P_{\mathrm{amt}}) \quad (1)$$

where $\alpha \geq 0$. However, standard CD needs two distinct models, increasing deployment complexity. Alternatively, DoLa (Chuang et al., 2023) contrasts layers within a single LLM, treating shallow layers as syntactic amateurs and deep ones as factual experts. Yet, DoLa is often ineffective on small models (Chuang et al., 2023) and incurs computational costs due to dynamic layer selection, particularly for large-vocabulary models like Qwen (Yang et al., 2025a).

**3.2. Motivation**

Our key insight is that MTP naturally encapsulates a hierarchy of "expert" and "amateur" distributions within a single model. Given the strong locality bias of LLMs (Liu et al., 2024; Khandelwal et al., 2018; Xiao et al., 2023; Press et al., 2021), the standard prediction $P(x_t|x_{<t})$ acts as a highly capable *Expert*, leveraging the full immediate context to resolve local nuances. In contrast, MTP predictions $P(x_t|x_{<t-k})$ operate without the recent $k$ tokens. These

delayed predictions serve as *Amateurs*, relying on global semantics or high-frequency patterns while lacking the information for specific local realization. By contrasting the locally precise Expert against these context-deficient Amateurs, we can effectively suppress generic background noise. This implies that the model's own past uncertainty can serve as an effective guide for its present decision-making.

**3.3. Temporal Guidance**

To materialize this insight, we propose **Temporal Guidance (TeGu)**, a decoding-time algorithm that ensembles predictions from different temporal offsets.

Let $\mathcal{M}$ be an LLM with the capability to predict future tokens. At decoding step $t$, the standard head produces the expert distribution $P_{\exp}(x_t) = P(x_t|x_{<t})$. Simultaneously, we retrieve cached last layer hidden states $h_{t-1-k}$, and utilize the MTP head corresponding to an offset of $k$ to compute the amateur distribution $P_{\mathrm{amt}}^{(k)}(x_t) = P(x_t|x_{<t-k})$. To generalize this, we employ a set of future offsets $\mathcal{K} = \{k_1, k_2, \ldots, k_m\}$ where each $k_i \geq 1$. We formulate the aggregated amateur distribution as a weighted mixture of predictions from these temporal offsets:

$$P_{\mathrm{amt}}(x_t) = \sum_{k \in \mathcal{K}} w_k P(x_t|x_{<t-k}), \quad (2)$$

where $w_k$ are manually configured hyperparameters serving as normalized weights ($\sum w_k = 1$) for each offset.

To ensure numerical stability during implementation, we compute the logarithm of this weighted sum using the LogSumExp (LSE) trick:

$$\log P_{\mathrm{amt}}(x_t) = \mathrm{LSE}_{k \in \mathcal{K}} \left( \log P(x_t|x_{<t-k}) + \log w_k \right). \quad (3)$$

where $\mathrm{LSE}(v_1, \ldots, v_n) = \log(\sum_i \exp(v_i))$. The final guided log-prob $\mathcal{V}(x_t)$ is obtained by applying the con-

trastive update:

$$\mathcal{V}(x_t) = \log P_{\exp}(x_t) + \alpha(\log P_{\exp}(x_t) - \log P_{\text{amt}}(x_t)). \quad (4)$$

From an information-theoretic perspective, this formulation implicitly maximizes the Conditional Pointwise Mutual Information between the specific local context and the target token. We provide a derivation in Appendix C, demonstrating how TeGu balances standard likelihood with local informativeness to suppress generic background noise.

Notably, a simplified variant of our method arises when we set $w_1 = 1$ (focusing solely on the immediate previous step), which we term **Bi-step Temporal Guidance**, which constitutes the primary focus of this work.

To mitigate the issue of plausible tokens being assigned zero probability caused by numerically unstable contrastive operations, we introduce the Adaptive Plausibility Constraint (APC) (Li et al., 2023). Conforming to the standard practice of Contrastive Decoding, we adopt an adaptive cutoff threshold $\tau = 0.1$, where tokens satisfying $P_{\exp}(x_t) < \tau \cdot \max_v P_{\exp}(v)$ are masked out during the decoding process. The inference flow of TeGu is shown in Figure 1 and Algorithm 1.

### 3.4. Conditional MTP Projector

To standardize the implementation of TeGu and ensure its universality across diverse model families, we introduce the **Conditional MTP Projector (cMTPP)**. Unlike naive MTP implementations that require independent heads for each offset (Gloeckle et al., 2024; Cai et al., 2024; Li et al., 2024), cMTPP employs a single lightweight shared module conditioned on the future offset $k$.

As illustrated in Figure 1, the projector utilizes adaptive modulation (adaLN) (Perez et al., 2018; Peebles & Xie, 2023) to dynamically process hidden states based on the step index. Crucially, to leverage the pre-trained semantic space and maintain parameter efficiency, we reuse the original frozen LM head ($W_{\text{LM}}$) of the base model. The probability distribution for a specific offset $k$ is computed as:

$$P(x_t|x_{<t-k}) = \text{Softmax}\left(W_{\text{LM}}\left(\mathbf{cMTPP}(h_{t-1-k}, k)\right)\right), \quad (5)$$

The detailed structure of Conditional MTP Projector is provided in Appendix A.1.

### 3.5. Training Objectives

We train the Conditional MTP Projector while keeping the backbone LLM frozen. A critical challenge involves the distribution shift between the projector's training data and the backbone's original pre-training corpus.

To mitigate this and ensure the amateur distribution aligns with the expert, we employ a hybrid objective combining

---

**Algorithm 1** Inference with Temporal Guidance

**Require:** Pre-trained LLM $\mathcal{M}$, Projector cMTPP, Context $x_{<t}$, Weights $\{w_k\}_{k \in \mathcal{K}}$, Contrast $\alpha$, Threshold $\tau$
**Ensure:** Generated token $x_t$
1: Let $K_{\max} = \max(\mathcal{K})$
2: **while** not EOS **do**
3:     Compute $h_{t-1} \leftarrow \mathcal{M}(x_{<t})$, and cache $h_{t-1}$
4:     $P_{\exp} \leftarrow \text{Softmax}(W_{\text{LM}}(h_{t-1}))$
5:     **for** each offset $k \in \mathcal{K}$ **do**
6:       $P_{\text{student}}^{(k)} \leftarrow \text{Softmax}(W_{\text{LM}}(\text{cMTPP}(h_{t-1-k}, k)))$
7:     **end for**
8:     $\log P_{\text{amt}} \leftarrow \text{LSE}_{k \in \mathcal{K}}\left(\log P_{\text{student}}^{(k)} + \log w_k\right)$
9:     $\mathcal{V} \leftarrow \text{APC}(\log P_{\exp} + \alpha(\log P_{\exp} - \log P_{\text{amt}}), \tau)$
10:     Sample $x_t \sim \text{Softmax}(\mathcal{V})$
11:     Update context $x_{<t+1} \leftarrow \{x_{<t}, x_t\}$, increment $t$
12:     Remove $h_{t-1-K_{\max}}$ from cache if exists
13: **end while**

---

Cross-Entropy (CE) loss and Knowledge Distillation (KD) (Hinton et al., 2015; Gu et al., 2023). The KD term minimizes the KL-divergence between the student's prediction and the frozen teacher's probability landscape. The total loss is formulated as a weighted linear combination across all time steps and future offsets:

$$\mathcal{L} = \sum_t \sum_{k \in \mathcal{K}} \lambda_k \left((1 - \lambda_{\text{KD}})\mathcal{L}_{\text{CE}}^{(t,k)} + \lambda_{\text{KD}}\mathcal{L}_{\text{KD}}^{(t,k)}\right), \quad (6)$$

where $\lambda_k$ is the weight assigned to the prediction at offset $k$. See Appendix A.2 for their specific definitions.

## 4. Experiments

### 4.1. Experimental Setup

**Training.** We conduct our training of the Conditional MTP Projector (cMTPP) using the `fineweb-edu` dataset (Penedo et al., 2024), where input sequences are truncated or padded to a length of 2048 tokens. We keep the entire LLM backbone and the original language model head completely frozen during training, exclusively optimizing the cMTPP module. The intermediate expansion ratio of SwiGLU FFN (Shazeer, 2020) is 2.7. The down-projection matrix of the projector is initialized to zero. We assign a weight of 0.3 to the cross-entropy loss and 0.7 to the distillation loss, with a distillation temperature set to 2.0. All experiments are implemented using the DeepSpeed framework (Rasley et al., 2020) with ZeRO-2 optimization and trained on a cluster of $8 \times$ NVIDIA RTX A5000 GPUs. For detailed settings, please refer to Appendix B.

**Baselines.** We compare with the following baselines: (1) Standard Contrastive Decoding (CD) (Li et al., 2023): using a smaller model from the same family as the expert model

*Table 1.* Main results across seven benchmarks evaluating mathematical reasoning (GSM8K, GSM8K-Platinum, Math500), code generation (HEval, HEval+, MBPP), and instruction following (IFEval). We compare the Baseline (Greedy) with advanced decoding strategies including standard Contrastive Decoding, DoLa, and our proposed **Temporal Guidance**. The bold values indicate the best result for each model on the respective benchmark and the values following $\pm$ represent the standard error.

| Model | Setting | GSM8K | GSM8K-Platinum | Math500 | HEval | HEval+ | MBPP | IFEval |
|---|---|---|---|---|---|---|---|---|
| **Qwen3-1.7B** | Baseline (Greedy) | $72.48_{\pm1.23}$ | $70.64_{\pm1.31}$ | $12.60_{\pm1.49}$ | $40.24_{\pm3.84}$ | $37.20_{\pm3.79}$ | $42.60_{\pm2.21}$ | $15.16_{\pm1.54}$ |
| | *Contrastive Decoding (Amateur Model Variations)* | | | | | | | |
| | Qwen3-0.6B | $73.31_{\pm1.22}$ | $71.29_{\pm1.28}$ | $11.40_{\pm1.44}$ | $39.02_{\pm3.18}$ | $35.98_{\pm3.29}$ | $43.60_{\pm2.10}$ | $20.33_{\pm1.89}$ |
| | *DoLa (Penalty Variations)* | | | | | | | |
| | Penalty 1.0 | $71.65_{\pm1.24}$ | $68.07_{\pm1.34}$ | $9.40_{\pm1.31}$ | $20.73_{\pm3.18}$ | $18.29_{\pm3.03}$ | $25.40_{\pm1.95}$ | $24.03_{\pm1.84}$ |
| | Penalty 1.2 | $72.93_{\pm1.22}$ | $71.05_{\pm1.30}$ | $9.20_{\pm1.29}$ | $28.05_{\pm3.52}$ | $25.00_{\pm3.39}$ | $24.80_{\pm1.93}$ | $26.06_{\pm1.89}$ |
| | *Temporal Guidance (Alpha Variations)* | | | | | | | |
| | Alpha 0.1 | $74.30_{\pm1.19}$ | $71.71_{\pm1.30}$ | $12.40_{\pm1.44}$ | $41.46_{\pm3.86}$ | $37.80_{\pm3.80}$ | $44.60_{\pm2.23}$ | $24.58_{\pm1.85}$ |
| | Alpha 0.2 | $\mathbf{75.51}_{\pm1.20}$ | $\mathbf{72.21}_{\pm1.28}$ | $\mathbf{12.80}_{\pm1.49}$ | $\mathbf{42.07}_{\pm3.82}$ | $\mathbf{38.41}_{\pm3.74}$ | $\mathbf{45.20}_{\pm2.23}$ | $\mathbf{26.99}_{\pm1.91}$ |
| **Llama-3.2-3B** | Baseline (Greedy) | $32.60_{\pm1.29}$ | $27.13_{\pm1.28}$ | $0.20_{\pm0.20}$ | $27.44_{\pm3.49}$ | $25.00_{\pm3.39}$ | $\mathbf{38.40}_{\pm2.18}$ | $6.65_{\pm1.07}$ |
| | *DoLa (Penalty Variations)* | | | | | | | |
| | Penalty 1.0 | $33.51_{\pm1.30}$ | $27.63_{\pm1.29}$ | $0.20_{\pm0.20}$ | $26.22_{\pm3.45}$ | $24.39_{\pm3.36}$ | $37.40_{\pm2.17}$ | $7.76_{\pm1.15}$ |
| | Penalty 1.2 | $30.10_{\pm1.26}$ | $26.88_{\pm1.28}$ | $0.00_{\pm0.00}$ | $29.27_{\pm3.56}$ | $27.44_{\pm3.49}$ | $36.20_{\pm2.15}$ | $8.32_{\pm1.19}$ |
| | *Temporal Guidance (Alpha Variations)* | | | | | | | |
| | Alpha 0.1 | $\mathbf{34.04}_{\pm1.29}$ | $\mathbf{28.37}_{\pm1.27}$ | $0.20_{\pm0.20}$ | $\mathbf{32.32}_{\pm3.66}$ | $\mathbf{29.27}_{\pm3.56}$ | $37.60_{\pm2.14}$ | $9.24_{\pm1.25}$ |
| | Alpha 0.2 | $32.75_{\pm1.25}$ | $26.39_{\pm1.22}$ | $0.00_{\pm0.00}$ | $31.71_{\pm3.64}$ | $28.05_{\pm3.52}$ | $37.00_{\pm2.16}$ | $\mathbf{10.17}_{\pm1.30}$ |
| **Qwen3-8B** | Baseline (Greedy) | $90.67_{\pm0.80}$ | $\mathbf{90.57}_{\pm0.84}$ | $20.40_{\pm1.80}$ | $63.41_{\pm3.77}$ | $58.54_{\pm3.86}$ | $64.80_{\pm2.14}$ | $24.77_{\pm1.86}$ |
| | *Contrastive Decoding (Amateur Model Variations)* | | | | | | | |
| | Qwen3-0.6B | $89.31_{\pm0.81}$ | $90.07_{\pm0.86}$ | $21.40_{\pm1.87}$ | $53.65_{\pm3.73}$ | $60.98_{\pm3.81}$ | $65.60_{\pm2.10}$ | $29.57_{\pm1.94}$ |
| | Qwen3-1.7B | $88.62_{\pm0.83}$ | $89.82_{\pm0.89}$ | $17.80_{\pm1.81}$ | $61.58_{\pm3.68}$ | $57.92_{\pm3.82}$ | $66.20_{\pm2.11}$ | $31.97_{\pm2.01}$ |
| | *DoLa (Penalty Variations)* | | | | | | | |
| | Penalty 1.0 | $89.84_{\pm0.83}$ | $90.16_{\pm0.86}$ | $19.40_{\pm1.77}$ | $66.46_{\pm3.70}$ | $60.98_{\pm3.82}$ | $65.00_{\pm2.14}$ | $25.88_{\pm1.88}$ |
| | Penalty 1.2 | $90.75_{\pm0.80}$ | $89.33_{\pm0.89}$ | $20.60_{\pm1.81}$ | $66.46_{\pm3.70}$ | $61.59_{\pm3.81}$ | $66.00_{\pm2.12}$ | $33.64_{\pm2.03}$ |
| | *Temporal Guidance (Alpha Variations)* | | | | | | | |
| | Alpha 0.1 | $90.37_{\pm0.81}$ | $89.83_{\pm0.87}$ | $22.20_{\pm1.86}$ | $\mathbf{68.29}_{\pm3.64}$ | $\mathbf{62.80}_{\pm3.79}$ | $65.60_{\pm2.13}$ | $24.77_{\pm1.86}$ |
| | Alpha 0.2 | $90.45_{\pm0.81}$ | $89.25_{\pm0.89}$ | $22.40_{\pm1.87}$ | $67.07_{\pm3.68}$ | $60.37_{\pm3.83}$ | $66.40_{\pm2.11}$ | $28.47_{\pm1.94}$ |
| | Alpha 0.3 | $90.67_{\pm0.80}$ | $89.83_{\pm0.87}$ | $22.20_{\pm1.86}$ | $64.63_{\pm3.74}$ | $59.76_{\pm3.84}$ | $\mathbf{67.20}_{\pm2.10}$ | $29.21_{\pm1.96}$ |
| | Alpha 0.4 | $\mathbf{91.05}_{\pm0.79}$ | $90.16_{\pm0.86}$ | $23.60_{\pm1.90}$ | $65.24_{\pm3.73}$ | $60.37_{\pm3.83}$ | $66.00_{\pm2.12}$ | $34.01_{\pm2.04}$ |
| | Alpha 0.5 | $90.52_{\pm0.81}$ | $89.83_{\pm0.87}$ | $\mathbf{24.20}_{\pm1.92}$ | $64.63_{\pm3.74}$ | $59.76_{\pm3.84}$ | $66.40_{\pm2.11}$ | $\mathbf{34.20}_{\pm2.04}$ |

as the amateur model. (2) DoLa: Decoding by Contrasting Layers (Chuang et al., 2023). It does not employ an additional model, but decodes by contrasting the probability distributions from the deep and shallow layers of the model. All implementations are based on the `custom_generate` function in Hugging Face Transformers (Wolf et al., 2020). Specifically, the DoLa implementation is taken directly from `transformers-community/dola`. Both standard CD and our TeGu are implemented by ourselves, also using the `custom_generate`.

**Models.** We employ models of different scales and series to verify the effectiveness and universality of our method. These include Qwen3-1.7B and Qwen3-8B from the Qwen series (Yang et al., 2025a), and Llama3.2-3B (Touvron et al., 2023) from the Llama series. For models that inherently support MTP, we employed MiMo-7B (Xiaomi et al., 2025).

**Evaluation.** We primarily employ three categories of benchmarks: mathematics, coding, and instruction following. The mathematics category comprises GSM8K (Cobbe et al., 2021), GSM8K-Platinum (Vendrow et al., 2025), and Math500 (Lightman et al., 2023). The coding category includes HEval (Chen, 2021), HEval+ (Liu et al., 2023), and MBPP (Austin et al., 2021). For instruction following, we utilize IFEval (Zhou et al., 2023). All evaluations are conducted in a standardized manner using the `lm-evaluation-harness` (Gao et al., 2024). Greedy sampling is employed for all methods throughout the generation process to ensure stability. For our TeGu, unless otherwise specified, all experiments are conducted by default using Bi-step Temporal Guidance.

### 4.2. Main Results.

**Outperforming Existing Methods.** As shown in Table 1, our proposed TeGu consistently outperforms the standard

*Table 2.* TeGu on MiMo-7B with its native MTP head.

| Alpha | GSM8K | GSM8K-P | HEval | HEval+ | IFEval | MBPP |
|---|---|---|---|---|---|---|
| 0.0 (Baseline) | 82.41 | 83.87 | 51.83 | 47.56 | 25.69 | **50.40** |
| 0.1 | **83.62** | **84.45** | 54.27 | 49.39 | 26.43 | 49.80 |
| 0.2 | 82.41 | 83.46 | 53.05 | 50.00 | **27.36** | 49.40 |
| 0.3 | 83.09 | 84.37 | **54.88** | **52.44** | 26.06 | 50.00 |

greedy decoding baseline across all model scales and the vast majority of task categories (Mathematics, Coding, and Instruction Following). Notably, on the Qwen3-1.7B model, TeGu achieves a significant improvement of 3.03% on GSM8K (75.51% vs. 72.48%) and 11.83% on IFEval (26.99% vs. 15.16%). This demonstrates that leveraging temporal information from the MTP head provides substantial guidance for accurate generation. TeGu also demonstrates superior performance compared to standard CD and DoLa. For instance, on Qwen3-8B, TeGu surpasses the best CD configuration on Math500 (24.20% vs. 21.40%) and IFEval (34.20% vs. 29.57%).

**Performance on Small Models.** We observe that DoLa exhibits significant instability on small-scale models like Qwen3-1.7B, suffering from decoding collapse even with a repetition penalty of 1.2. For example, its performance drastically drops from 42.60% to 24.80% on MBPP and from 40.24% to 28.05% on HEval. This failure suggests that the layer-contrastive assumption breaks down in smaller models due to entangled representations. In contrast, TeGu demonstrates strong robustness, consistently outperforming baselines without auxiliary penalties, confirming that temporal contrast provides a more universal guidance signal across different model scales.

**TeGu on LLM with Native MTP.** We also experiment with models that natively support MTP. We select MiMo-7B (Xiaomi et al., 2025), which comes equipped with an MTP head capable of predicting the next-next token. As shown in Table 2, TeGu effectively activates the guidance capabilities of MiMo's MTP head. TeGu boosts performance on HEval from 51.83% to 54.88% and HEval+ from 47.56% to 52.44%. These demonstrate that TeGu is not tied to cMTPP. For LLMs with native MTP heads, TeGu can be directly deployed as a generalized, training-free method.

### 4.3. Analysis on TruthfulQA

As shown in Table 3, TeGu exhibits largely neutral effects on the TruthfulQA benchmark (Lin et al., 2022). Across both MC1 and MC2 tasks, TeGu achieves performance comparable to greedy decoding on Qwen3-8B and Llama-3.2-3B, indicating limited impact on factual accuracy. In contrast, DoLa substantially improves MC1 accuracy but causes a severe degradation in MC2 performance, suggesting over-specialization toward narrow factual cues at the expense of overall generation quality.

*Table 3.* Performance comparison on TruthfulQA across different models. TeGu maintains performance comparable to the baseline.

| Model | Method | MC1 Accuracy (%) | MC2 Score (%) |
|---|---|---|---|
| Llama-3.2-3B | Greedy (Base) | 18.24 | 74.29 |
| | DoLa | **54.47** | 42.09 |
| | TeGu ($\alpha = 0.1$) | 18.24 | **74.30** |
| | TeGu ($\alpha = 0.2$) | 18.24 | 74.29 |
| | TeGu ($\alpha = 0.3$) | 18.12 | 74.21 |
| Qwen3-8B | Greedy (Base) | 28.15 | 80.60 |
| | DoLa | **88.00** | 17.38 |
| | TeGu ($\alpha = 0.1$) | 28.64 | **80.65** |
| | TeGu ($\alpha = 0.2$) | 28.40 | 80.61 |
| | TeGu ($\alpha = 0.3$) | 28.27 | 80.43 |

*Table 4.* Performance comparison on Wikitext-2 using Qwen3-8B, reported via Distinct-n and Rep-4 Rate.

| Method | $\alpha$ | Distinct-1 (%) ↑ | Distinct-2 (%) ↑ | Rep-4 Rate (%) ↓ |
|---|---|---|---|---|
| Greedy | - | 11.02 | 32.76 | 35.84 |
| DoLa | - | 11.48 | 36.83 | 30.35 |
| **TeGu** | 0.1 | 11.32 | 33.79 | 33.70 |
| **TeGu** | 0.2 | 12.66 | 39.37 | 25.33 |
| **TeGu** | 0.3 | **13.20** | **41.66** | **20.43** |

These results reflect the differing objectives of contrastive decoding strategies. Layer-wise contrast methods such as DoLa are more effective for isolating factual signals and reducing certain hallucinations, whereas TeGu, based on temporal prediction contrast, primarily benefits long-horizon reasoning and coherence without modifying factual knowledge retrieval. Accordingly, TeGu is better suited for reasoning-, coding-, and instruction-oriented tasks rather than factuality-focused benchmarks like TruthfulQA.

### 4.4. Repetition Analysis

Table 4 reports the evaluation results on Wikitext-2 (Merity et al., 2016) using Qwen3-8B. We compare our proposed TeGu with standard greedy decoding and DoLa. Deterministic decoding exhibits severe degeneration with a high Rep-4 Rate of 35.84%. Although DoLa lowers the Rep-4 Rate to 30.35%, Appendix D shows that it still struggles with recursive phrases in long contexts.

TeGu outperforms baselines in mitigating repetition and improving diversity. With $\alpha = 0.3$, TeGu achieves a Rep-4 Rate of 20.43%, which is a reduction of 43% over the Base model and 32.7% over DoLa. Concurrently, the Distinct-2 score improves to 41.66%. Analysis in Appendix D also confirms that TeGu effectively suppresses repetitive tokens.

## 5. Extensive Analysis

### 5.1. Impact of Training Objectives

To validate the necessity of the KL loss discussed in Section 3, we conduct an ablation study on the training objectives of cMTPP. We compare the performance of TeGu when the

*Table 5.* Ablation study on amateur distribution selection: single vs. weighted multi-temporal offset performance on Qwen3-8B. Avg Acc represents the average accuracy across all benchmarks.

| Settings | Amateur Weights | | | Avg Acc |
|---|---|---|---|---|
| | $k=2$ | $k=3$ | $k=4$ | |
| Greedy (Base) | 0.00 | 0.00 | 0.00 | 59.91 |
| 1 Amateur | 1.00 | 0.00 | 0.00 | **61.49** |
| | 0.00 | 1.00 | 0.00 | 60.73 |
| | 0.00 | 0.00 | 1.00 | 60.21 |
| 2 Amateurs | 0.30 | 0.70 | 0.00 | 61.05 |
| | 0.50 | 0.50 | 0.00 | 60.87 |
| | 0.70 | 0.30 | 0.00 | 60.69 |
| | 0.00 | 0.30 | 0.70 | 59.30 |
| | 0.00 | 0.50 | 0.50 | 60.36 |
| | 0.00 | 0.70 | 0.30 | 60.52 |
| 3 Amateurs | 0.50 | 0.30 | 0.20 | 61.24 |
| | 0.30 | 0.50 | 0.20 | 61.07 |
| | 0.20 | 0.30 | 0.50 | 61.16 |
| | 0.34 | 0.33 | 0.33 | 61.32 |

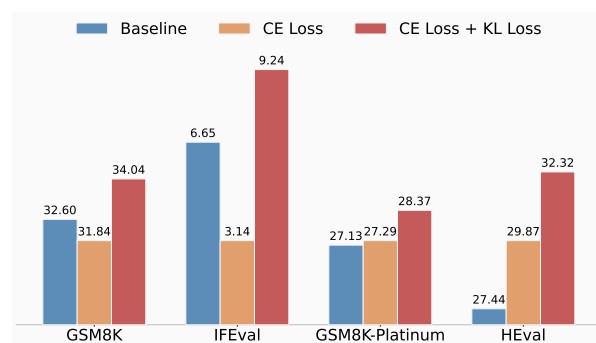

*Figure 2.* Ablation study on training objectives. We evaluate TeGu on Llama-3.2-3B across four benchmarks. The results compare the Baseline against the MTP projector trained with only Cross-Entropy loss (CE Loss) and our proposed method combining CE and KL divergence (CE Loss + KL Loss).

projector is trained solely with the Cross-Entropy loss ($\mathcal{L}_{\text{CE}}$) versus the combined objective ($\mathcal{L}_{\text{CE}} + \mathcal{L}_{\text{KD}}$).

As shown in Figure 2, optimizing the MTP projector using only CE Loss yields unstable performance. While it achieves marginal gains on HEval, it significantly degrades performance on IFEval and GSM8K compared to the Baseline. This empirical evidence supports our motivation: due to the lack of the original pre-training corpus, relying solely on CE loss causes the MTP head to overfit the specific post-training data. This results in a distribution shift where the amateur becomes misaligned with the expert, leading to harmful contrastive guidance. After incorporating the KL loss, TeGu achieved improvements across all benchmarks and consistently outperformed the model trained solely with CE loss. This demonstrates that aligning the distribution of MTP with that of NTP is crucial for TeGu.

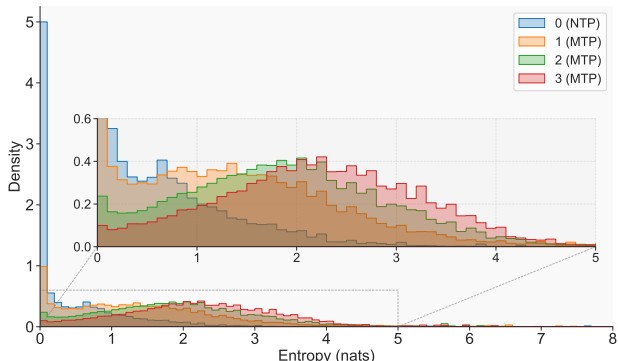

*Figure 3.* The distribution entropy density histograms of NTP and MTP heads for the Qwen3-8B model on the MATH500.

### 5.2. Entropy Analysis

To empirically validate the hypothesis that Multi-Token Prediction modules can serve as effective amateur models, we analyze the uncertainty of predictions generated by different heads. We compute the entropy of the probability distributions produced by the standard Next Token Prediction head and the auxiliary heads using the MATH500 dataset (Lightman et al., 2023). This metric serves as a proxy for model confidence where lower entropy signifies higher certainty.

As illustrated in Figure 3, there is a distinct hierarchical pattern in the distributional properties. The standard head at offset 0 exhibits a density distribution that is heavily skewed towards zero entropy. This sharpness indicates that the model possesses high confidence in its predictions when given access to the full immediate context, thereby acting as a capable expert. Conversely, the auxiliary heads corresponding to future offsets display a clear trend where the entropy distribution becomes flatter and shifts towards higher values. This increase in entropy reflects the growing uncertainty as the model attempts to predict tokens without observing the immediate preceding context. The observable gap between the low-entropy expert and the high-entropy amateurs provides the necessary statistical foundation for our proposed Temporal Guidance method to effectively distinguish and amplify context-specific signals.

### 5.3. Impact of Amateur Selection

We investigate the influence of temporal offsets $\mathcal{K}$ and aggregation weights $w_k$. Table 5 summarizes the results regarding single versus ensemble amateur configurations.

**Effect of Temporal Distance.** Performance degrades as the temporal distance $k$ increases. The nearest offset ($k=1$) achieves the highest accuracy of 61.49%, whereas more distant states ($k=2$ and $k=3$) drop to 60.73% and 60.21%, respectively. This indicates that immediate past states retain the most relevant context for contrastive decoding.

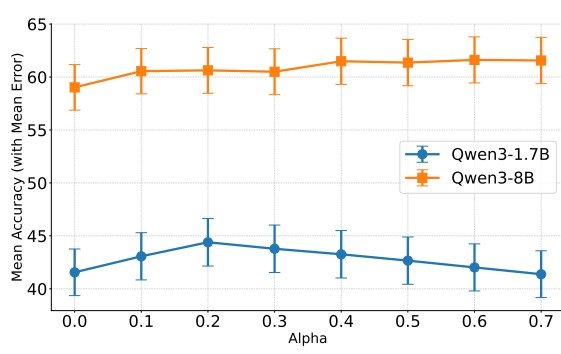

*Figure 4.* Ablation study on the hyperparameter $\alpha$. The figure reports the Mean Accuracy of Qwen3-1.7B and Qwen3-8B models under different $\alpha$ values. Error bars indicate the standard error.

**Single vs. Mixture Amateurs.** Aggregating distributions from multiple offsets fails to outperform the single best configuration. As shown in Table 5, mixtures (e.g., combining $k = 1$ and $k = 2$) yield lower accuracies ($< 61.05\%$) than using $k = 1$ alone. This suggests that weighted ensembles dilute the strong contrastive signal provided by the most informative historical state.

These findings validate the design of our Bi-step Temporal Guidance. Relying on a single, proximal offset is not only more computationally efficient, but also more effective than complex ensembles.

### 5.4. Analysis of $\alpha$

To investigate the impact of the guidance strength hyperparameter $\alpha$ on the performance of TeGu, we conducted ablation experiments on Qwen3-1.7B and Qwen3-8B. Figure 4 illustrates the trends in mean accuracy of all tasks across varying $\alpha$ settings.

The results reveal scale-dependent behavior regarding the choice of $\alpha$. The smaller Qwen3-1.7B model exhibits higher sensitivity to the contrastive penalty; its performance peaks at $\alpha = 0.2$ but degrades slowly as the penalty strength increases. This may be attributed to the inherent capacity limitations of small models. For smaller models, a conservative setting (keeping $\alpha < 0.3$) is preferable to prevent over-penalization of the probability distribution.

In contrast, the larger Qwen3-8B model demonstrates remarkable robustness. Its performance consistently improves or maintains high stability as $\alpha$ increases, even reaching $\alpha = 0.7$ without signs of degradation. This indicates that larger models, with their more refined representations, are capable of tolerating and effectively leveraging stronger contrastive signals to suppress amateur errors.

### 5.5. Efficiency Analysis

We evaluate the computational efficiency of TeGu in comparison with greedy decoding, standard CD, and DoLa. De-

*Table 6.* Comparison of inference VRAM usage (GB) and total elapsed time (seconds) on MATH500 and HumanEval benchmark. CD denotes standard Contrastive Decoding.

| Method | Math500 | | | HumanEval | | |
|---|---|---|---|---|---|---|
| | Acc. | Mem | Time | Acc. | Mem | Time |
| *Backbone: Qwen3-1.7B* | | | | | | |
| Greedy Decoding | 12.60 | 4.82 | 634 | 40.24 | 5.64 | 887 |
| CD (w/ Qwen3-0.6B) | 11.40 | 8.44 | 1273 | 39.02 | 8.48 | 1486 |
| DoLa | 9.40 | 5.47 | 769 | 28.05 | 5.95 | 1069 |
| **TeGu (Ours)** | **12.80** | **5.53** | **648** | **42.07** | **5.72** | **927** |
| *Backbone: Qwen3-8B* | | | | | | |
| Greedy Decoding | 20.40 | 17.72 | 835 | 63.41 | 19.08 | 1247 |
| CD (w/ Qwen3-0.6B) | 21.40 | 20.96 | 1517 | 53.65 | 20.95 | 2077 |
| CD (w/ Qwen3-1.7B) | 17.80 | 23.11 | 1544 | 61.58 | 22.97 | 2036 |
| DoLa | 20.60 | 19.22 | 1149 | 66.46 | 19.81 | 1634 |
| **TeGu (Ours)** | **24.20** | **19.72** | **960** | **68.29** | **19.53** | **1326** |

tailed metrics, including memory usage and inference time, are summarized in Table 6.

**Memory Overhead.** Standard CD significantly exacerbates memory consumption by requiring a separate amateur model. For instance, pairing Qwen3-8B with a 1.7B amateur increases VRAM usage by ~30% (17.72 GB to 23.11 GB). In contrast, TeGu avoids external models by leveraging the internal Conditional MTP Projector and cached historical states. Consequently, it maintains a memory footprint comparable to the base model with only marginal overhead, ensuring suitability for memory-constrained environments.

**Inference Latency.** Standard CD nearly doubles inference time due to dual forward passes, while DoLa incurs noticeable costs from multi-layer logit contrasts. TeGu minimizes redundant computation by reusing the MTP output from the previous step. As a result, TeGu is significantly faster than both baselines, incurring only a minor latency increase (2%~15%) over the base model while delivering substantial performance gains.

## 6. Conclusion

In this work, we proposed **Temporal Guidance (TeGu)**, a novel decoding framework that leverages the temporal dimension of LLMs to enhance generation quality. By contrasting the standard predictions with context-deficient amateur signals derived from our minimalist Conditional Multi-Token Prediction Projector (cMTPP), TeGu effectively suppresses generic noise and repetition without requiring external models. Comprehensive experiments demonstrate that TeGu outperforms standard greedy decoding, standard CD and DoLa across mathematics, coding, and instruction-following benchmarks. Notably, TeGu overcomes the instability of layer-wise contrasts on smaller-scale models while maintaining negligible memory and latency overhead, offering a practical solution for high-quality text generation.

## Impact Statement

This paper presents work whose goal is to advance the field of Machine Learning. There are many potential societal consequences of our work, none which we feel must be specifically highlighted here.

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

## A. Details of Conditional MTP Projector

### A.1. Conditional MTP Projector Architecture

In the main text, we introduced the projection module $\textbf{cMTPP}(h, k)$. Here we detail its internal computation. Let $h \in \mathbb{R}^d$ be the hidden state. We first embed the step index $k$ into a vector $e_k$ and employ adaLN (Perez et al., 2018; Peebles & Xie, 2023) to modulate the normalized hidden state:

$$\gamma, \beta = \text{adaLN}(e_k),$$
$$\tilde{h} = \text{RMSNorm}(h) \odot (1 + \gamma) + \beta, \tag{7}$$

where $\odot$ denotes element-wise multiplication. The modulated vector is then processed by a gated MLP (Shazeer, 2020):

$$\textbf{cMTPP}(h, k) = W_{\text{down}} \left( \sigma(W_{\text{gate}}\tilde{h}) \odot W_{\text{up}}\tilde{h} \right). \tag{8}$$

### A.2. Loss Function Definitions

The total loss $\mathcal{L}$ combines two components. For a target token $x_t$ and offset $k$, the standard Cross-Entropy loss is:

$$\mathcal{L}_{\text{CE}}^{(t,k)} = -\log P_{\text{student}}(x_t | x_{<t-k}). \tag{9}$$

To transfer the base model's generalization capability, the Knowledge Distillation loss is defined as the KL-divergence between the teacher (expert) and student (amateur) distributions:

$$\mathcal{L}_{\text{KD}}^{(t,k)} = D_{\text{KL}} \left( P_{\text{teacher}}(\cdot | x_{<t}) \| P_{\text{student}}(\cdot | x_{<t-k}) \right). \tag{10}$$

## B. Implementation Details and Hyperparameters

We utilize the AdamW optimizer (Loshchilov & Hutter, 2017) for training the Conditional MTP Projector. The training process is conducted using BFloat16 precision to optimize memory usage and computational efficiency. We employ a cosine annealing learning rate schedule (Loshchilov & Hutter, 2016) with a linear warmup phase. All training can be completed within a few hours on $8 \times$ NVIDIA RTX A5000 GPUs. The specific hyperparameters used across our experiments are summarized in Table 7.

*Table 7.* Hyperparameters for training the Conditional MTP Head.

| Hyperparameter | Value |
|---|---|
| Dataset | HuggingFaceFW/fineweb-edu |
| Sequence Length | 2048 |
| Global Batch Size | 128 (16 × 8 GPUs) |
| Total Training Steps | 3000 |
| Optimizer | AdamW |
| Peak Learning Rate | $2.0 \times 10^{-4}$ |
| LR Schedule | Cosine Annealing |
| Warmup Ratio | 0.05 (5%) |
| Precision | BFloat16 |
| MTP Projector Expansion Ratio | 2.7 |
| Loss Weight ($\lambda_{CE}$) | 0.3 |
| Loss Weight ($\lambda_{KD}$) | 0.7 |
| Distillation Temperature ($T$) | 2.0 |

## C. Theoretical Analysis of Temporal Guidance

In this section, we provide a theoretical interpretation of the proposed Temporal Guidance (TeGu) method from an information-theoretic perspective. We demonstrate that our contrastive formulation implicitly maximizes the Conditional Pointwise Mutual Information (CPMI) between the immediate local context and the target token, given the distant past.

### C.1. Problem Formulation

Let the full context sequence at step $t$ be denoted as $C_{\text{full}} = x_{<t}$. Given that LLMs typically prioritize the immediate local tokens when generating text (Liu et al., 2024; Khandelwal et al., 2018; Xiao et al., 2023; Press et al., 2021), this insight motivates us to decompose this context into two partitions:

- **Distant Past ($C_{\text{past}}$):** The sequence $x_{<t-k}$, representing global background and high-level semantics.

- **Local Context ($C_{\text{local}}$):** The sequence of recent tokens $\{x_{t-k}, \ldots, x_{t-1}\}$, which contains immediate syntactic cues and specific triggers for the next token.

Thus, we have $C_{\text{full}} = \{C_{\text{past}}, C_{\text{local}}\}$. The standard autoregressive generation (the Expert) models the likelihood conditioned on the full context:

$$P_{\text{exp}}(x_t) = P(x_t \mid C_{\text{full}}) = P(x_t \mid C_{\text{past}}, C_{\text{local}}). \tag{11}$$

The aggregated Amateur distribution, derived from Multi-Token Prediction heads with temporal offsets, effectively approximates the likelihood conditioned primarily on the distant past:

$$P_{\text{amt}}(x_t) \approx P(x_t \mid C_{\text{past}}). \tag{12}$$

### C.2. Connection to Mutual Information

A known issue in likelihood maximization is the bias towards generic, high-frequency tokens (Holtzman et al., 2019a). To mitigate this, we seek a token $x_t$ that is specifically informative regarding the Local Context $C_{\text{local}}$. We quantify this specificity using the Pointwise Mutual Information (PMI) between the next token $x_t$ and the local context (Li et al., 2016), conditioned on the shared past:

$$\text{PMI}(x_t; C_{\text{local}} \mid C_{\text{past}}) = \log \frac{P(x_t, C_{\text{local}} \mid C_{\text{past}})}{P(x_t \mid C_{\text{past}})P(C_{\text{local}} \mid C_{\text{past}})}. \tag{13}$$

Applying the chain rule of probability to the joint distribution in the numerator, we have $P(x_t, C_{\text{local}} \mid C_{\text{past}}) = P(x_t \mid C_{\text{local}}, C_{\text{past}})P(C_{\text{local}} \mid C_{\text{past}})$. Substituting this back into the PMI equation simplifies it as follows:

$$
\begin{aligned}
\text{PMI}(x_t; C_{\text{local}} \mid C_{\text{past}}) &= \log \frac{P(x_t \mid C_{\text{local}}, C_{\text{past}})P(C_{\text{local}} \mid C_{\text{past}})}{P(x_t \mid C_{\text{past}})P(C_{\text{local}} \mid C_{\text{past}})} \\
&= \log P(x_t \mid C_{\text{full}}) - \log P(x_t \mid C_{\text{past}}) \\
&= \log P_{\text{exp}}(x_t) - \log P_{\text{amt}}(x_t).
\end{aligned}
\tag{14}
$$

Equation 14 reveals that the difference between the expert and amateur logits mathematically represents the information gain provided specifically by the local context $C_{\text{local}}$.

### C.3. Decoding Objective

An ideal decoding strategy balances fluency (high likelihood) and informativeness (high specificity). We formulate the scoring function $\mathcal{V}(x_t)$ as a linear combination of the standard log-likelihood and the PMI term:

$$
\begin{aligned}
\mathcal{V}(x_t) &= \log P_{\text{exp}}(x_t) + \alpha \cdot \text{PMI}(x_t; C_{\text{local}} \mid C_{\text{past}}) \\
&= \log P_{\text{exp}}(x_t) + \alpha \left(\log P_{\text{exp}}(x_t) - \log P_{\text{amt}}(x_t)\right).
\end{aligned}
\tag{15}
$$

This derivation aligns perfectly with the Temporal Guidance update rule. The hyperparameter $\alpha$ controls the weight of the local context's specific contribution, effectively suppressing hallucinations or generic continuations that arise solely from the distant past.

## D. Qualitative Analysis

To explicitly demonstrate the effectiveness of our method in mitigating degeneration, we provide generated samples from the Wikitext-2 test set using the prompt: "*...003 . He has also appeared in the television series Doctors and Holby City* .". The comparison includes the Base model (Greedy Search), DoLa, and TeGu ($\alpha = 0.3$).

**Base Model.** As shown in Table 8, the standard autoregressive model quickly falls into a repetition loop. After generating the first sentence, it continuously repeats the phrase "He has also appeared in the film The Last King of Scotland" without introducing any new information.

**DoLa.** Table 9 shows that DoLa attempts to alleviate the issue by alternating between two repeating patterns ("Waterloo Road and Doctors" and "The Last King of Scotland"). Although it exhibits slightly more variation than the Base model, it fails to break away from the local repetitive cycle.

**TeGu (Ours).** In contrast, as displayed in Table 10, TeGu ($\alpha = 0.3$) generates a plausible and diverse continuation. While the sentence structure remains consistent (which is typical for biographical entries in this dataset), the model successfully retrieves distinct entity names such as "The Golden Compass," "The Da Vinci Code," "Midsomer Murders," and "Silent Witness." This confirms that our Temporal Guidance mechanism effectively penalizes the repetitive tokens favored by the amateur distribution, thereby forcing the model to explore more informative paths.

*Table 8.* Generation sample from the **Base** model (Greedy).

| |
| --- |
| **Output:** ...He has also appeared in the film The Last King of Scotland . He has also appeared in the film The Last King of Scotland . He has also appeared in the film The Last King of Scotland . He has also appeared in the film The Last King of Scotland . He has also appeared in the film The Last King of Scotland ... |

*Table 9.* Generation sample from **DoLa**.

| |
| --- |
| **Output:** ...He has also appeared in the film The Last King of Scotland . He has also appeared in the television series Waterloo Road and Doctors . He has also appeared in the film The Last King of Scotland . He has also appeared in the television series Waterloo Road and Doctors . He has also appeared in the film The Last King of Scotland ... |

*Table 10.* Generation sample from **TeGu** ($\alpha = 0.3$).

| |
| --- |
| **Output:** ...He has also appeared in the film The Golden Compass . He has also appeared in the film The Da Vinci Code . He has also appeared in the film The Constant Gardener . He has also appeared in the film The Golden Compass . He has also appeared in the television series Midsomer Murders . He has also appeared in the television series Silent Witness ... |

## E. Limitations

Despite the significant improvements in reasoning and generation quality demonstrated by Temporal Guidance, we acknowledge a few limitations in our current work:

**Limited Impact on Factuality Benchmarks.**   While our method effectively enhances performance on reasoning-intensive tasks and reduces repetition, its impact on pure factuality benchmarks remains neutral. Our experiments indicate that while temporal contrast improves coherence and logic, it may not effectively isolate factual knowledge retrieval mechanisms from hallucinations in the same way layer-wise contrasting does.

**Training Overhead for Non-MTP Models.**   Although TeGu can be deployed as a training-free decoding strategy for models with native Multi-Token Prediction capabilities, the majority of current open-source LLMs lack these auxiliary heads. To apply our method to these standard architectures, we must train a Conditional MTP Projector. This module introduces an additional training stage compared to strictly inference-time methods.

**Hyperparameter Sensitivity on Smaller Models.**   Our ablation studies indicate that the robustness of TeGu regarding the guidance strength is scale-dependent. While larger models remain stable across a wide range of values, smaller models exhibit higher sensitivity. Therefore, for small models, a smaller value of $\alpha$ is sufficient.

## F. The Use of Large Language Models

In the preparation of this manuscript, a Large Language Model (LLM) was used solely for the purpose of language polishing and stylistic refinement of the text. The LLM was prompted to improve clarity, grammar, and fluency of expression, without altering the core scientific content, methodology, results, or interpretations presented in the paper. The research ideas, experimental design, data analysis, and original writing were entirely conducted by the human authors. The LLM did not contribute to the generation of hypotheses, formulation of research questions, or development of novel concepts. Its role was strictly limited to post-writing linguistic enhancement.

