# OpenReview forum: "Temporal Guidance for Large Language Models"
_ICML.cc/2026/Conference — Submitted to ICML 2026_

### Official Review · Reviewer_9aAb · 2026-02-22

**Soundness:** 1
**Presentation:** 3
**Significance:** 1
**Originality:** 2
**Overall Recommendation:** 2
**Confidence:** 5

**Summary:**

This paper propose TeGu (Temporal Guidance), a new decoding approach that exploits the temporal dimension of LLMs. The key idea is that LM predictions without access to the most recent context may serve as good “amateur” guesses that can be contrastively factored out when computing the next-token probability distribution. By incorporating historical predictions from MTP (multi-token prediction) modules as “amateurs,” it operates as a form of self-contrastive decoding (using just 1 model). The authors also introduce cMTPP, a lightweight projector training framework that allows TeGu to work with off-the-shelf LMs that do not have native MTP support. cMTPP is trained on top of the backbone LLM using both a cross-entropy and soft knowledge distillation loss on a standard pre-training corpus.

 TeGU outperforms baselines (e.g., vanilla greedy decoding, DoLA, contrastive decoding) on math, coding, and instruction-following benchmarks, across 3 different base models. TeGU is most effective for long-horizon constrained generation tasks (e.g., IFEval), and less so for tasks such as factuality. Besides main results, the authors also provide extensive ablations and analysis to justify their methodology.

**Compliance With Llm Reviewing Policy:**

Affirmed.

**Final Justification:**

I am keeping my original score, mainly due to the soundness consideration. The authors' rebuttal and subsequent responses did not fully address my main concerns, and we may be somewhat talking past each other.

**Key Questions For Authors:**

1. The choice of hyper parameters used for cMTPP training feels arbitrary to me (Section 4.1). Can the authors please provide more information on how these hyper parameters, in particular, the loss weights, were selected?
2. In Table 1, what is the standard error calculated over? I.e., are you reporting average performance + SE over 3 random seeds?
How much parameter overhead does cMTPP introduce? Does the overhead scale with model size?
3. I understand why you chose base models / greedy decoding for experiments, but in reality, people mainly decode with instruct models in sampling regimes. It might be worth exploring these decoding adjustments somewhere in your paper.

Formatting comments:
1. The citation formatting is off in some places (i.e., it should be ordered first by year of publication, then alphabetically by name).
2. There are some phrases here are there (e.g., ingenious, superior performance) that feel too much like self-praise and feel tonally out of place for a scientific report — I would recommend removing them entirely for the next draft.
3. Some details (e.g., logsumexp trick) are quite standard and can be relegated to the appendix.
The bolding in Table 6 is misleading…bolding results typically denote the best (i.e., either the largest or smallest) values, but this is not true for the Mem or Time columns.

**Limitations:**

yes

**Strengths And Weaknesses:**

Soundness: Conceptually, the proposed methodology makes sense to me, and I think the entropy analysis in 5.2 is a good analytical justification. I can buy that TeGu is particularly well-suited for long-horizon tasks but not on instant recall tasks (e.g., factuality), and I commend the authors for reporting negative results on TruthfulQA in 5.3. However, looking at Table 1, I for not find the reported results to match the strength of the claims. If you compare TeGu at the best alpha against the simplest greedy baseline, the results are very similar (and in many cases, not statistically significant) for practically all the benchmarks except IFEval.  And for IFEval, it appears that the other baselines (CD and DoLA) are competitive. While these results can be attributed to the cMTPP not being very well-trained, if we look at Table 2, MiMo-7B (which has a native MTP head) also shows very negligible performance improvement compared to vanilla greedy decoding, except for on HEval/HEval++. If we look at the qualitative TeGu decoding example in Table 10, the output is still qualitatively degenerate; despite showing less repetition, it still follows a repetitive generation loop (likely as you used greedy decoding throughout). It’s hard for me to support this method when the greedy baseline seems pretty competitive and does not incur a memory / efficiency overhead, but I’m happy to be convinced otherwise.

Presentation: For the most part, I find the paper easy to follow (I’ve also left some presentation comments for the authors that are easy to fix) and the authors provided enough detail to replicate the paper, along with their anonymized code. Sections 3.4-3.5 (introducing cMTPP) was personally a little hard to get through; instead of providing details on what they did, maybe the authors can consider spending more time justifying particular design choices.

Significance: This is pretty subjective, but I personally feel the contribution of this work is incremental (and the results are not convincing enough for me to lean toward acceptance). The idea of decoding with experts / amateurs is not new, nor is multi-token prediction as a concept. TeGu is not straightforward to adapt to off-the-shelf LLMs and requires a dedicated training process for a new prediction module. I find the argument that TeGu works for long-horizon (constrained?) tasks interesting, though, maybe the authors can consider pushing harder along this direction.

Originality: This paper offers a methodological contribution that offers a novel combination of existing techniques, and the authors do a nice job of situating it in relevant work. I think the most impressive component is that the authors show that TeGu can be retrofitted to LMs without native MTP support. However, the originality of this contribution is blunted by the empirical results; please see my comments above.

---

> ### Author Rebuttal · Authors · 2026-03-30
>
> We sincerely thank you for your thorough review and for recognizing the theoretical soundness of our entropy analysis, as well as the value of our zero-shot adaptation of MTP for standard LLMs. We have carefully addressed your concerns, conducting new experiments to strengthen the empirical evidence of our paper.
>
> ---
> > However, looking at Table 1, I for not find the reported results to match the strength of the claims...
> > In Table 1, what is the standard error calculated over? I.e., are you reporting average performance + SE over 3 random seeds?
>
> We apologize for the confusion regarding the standard errors (SE) in Table 1. Because greedy decoding is deterministic, these SEs are not variance over random seeds. They are binomial standard errors automatically reported by `lm-evaluation-harness`, computed as $\sqrt{p(1-p)/N}$ (where $p$ is accuracy and $N$ is sample size).
>
> Overlapping binomial confidence intervals do not imply insignificance for paired data. We are evaluating the performance difference $D=X_{TeGu}-X_{Base}$. The variance of this difference is:
>
> $$
> \operatorname{Var}(D)=\operatorname{Var}(X_{TeGu})+\operatorname{Var}(X_{Base})-2\operatorname{Cov}(X_{TeGu},X_{Base})
> $$
> Since the paired outputs under different decoding strategies are highly correlated, the covariance term is large, which significantly reduces the true variance of the difference.
>
> To rigorously validate our claims, we conducted an exact McNemar's test on the paired predictions. The $p$-values below confirm TeGu's improvements are statistically significant ($p<0.05$) in most key scenarios. The few $p>0.05$ cases primarily result from baseline saturation or small benchmark sample sizes (e.g., HEval) that limit statistical power:
>
> Model Name|Gsm8k|Gsm8k-P|Math500|HEval|HEval+|Mbpp|IFEval
> -|-|-|-|-|-|-|-
> qwen3-1.7b|0.0072|0.0344|1|0.25|0.5|0.0241|6.347e-06
> llama3.2-3b|0.0319|0.0721|1|0.0385|0.0654|0.4239|0.010973
> qwen3-8b|0.5113|0.6029|0.0183|0.0214|0.0390|0.0226|0.000107
>
> > I understand why you chose base models / greedy decoding for experiments, but in reality, people mainly decode with instruct models in sampling regimes...
>
> We agree with your perspective. Accordingly, we have supplemented our experiments with Qwen3-1.7B using sampling-based generation:
>
> Alpha|Gsm8k|Humaneval|Mbpp
> -|-|-|-
> 0.0 (baseline) |72.80±0.75|33.69±1.53|41.50±1.39
> 0.1|72.99±0.63|35.67±2.36|42.40±1.94
> 0.2|74.53±0.98|36.89±2.02|45.00±1.26
> 0.3|72.90±0.36|36.13±2.65|42.35±1.31
>
> The generation parameters were set to top-$p=0.95$, top-$k=50$, and temperature$=1.0$, with 4 sampling runs conducted to compute the mean and standard deviation. As observed, TeGu enhances model performance even under sampling-based decoding, further substantiating its effectiveness.
>
> >  If we look at the qualitative TeGu decoding example in Table 10, the output is still qualitatively degenerate...
>
> The repetitive sentence structure ("He has also appeared in...") is largely a stylistic artifact of the Wikitext-2 biographical data. The critical improvement lies in entity generation. While the Baseline and DoLa collapse into an infinite token-level loop (repeating "The Last King of Scotland"), TeGu successfully breaks this degeneration by retrieving diverse, distinct entities ("The Golden Compass," "The Da Vinci Code"). TeGu forces informative exploration despite rigid syntax.
>
> > The choice of hyper parameters used for cMTPP training...
>
> We use a hybrid objective because combining CE and KL losses accelerates convergence and yields slightly better final performance than KD-only training. The hard labels from the CE loss provide more explicit gradients.
>
> Regarding the mixing parameter, $\lambda_{KD} = 0.7$ was chosen as an effective empirical value. Our primary goal is to align the amateur distribution with the original model (hence the larger weight on KD), while utilizing CE to speed up training. This configuration already delivers highly satisfactory results without the need for exhaustive hyperparameter search.
>
> > How much parameter overhead does cMTPP introduce? Does the overhead scale with model size?
>
> The primary parameters of cMTPP originate from the SwiGLU FFN, scaling approximately as $10 d_{\text{model}}^2$ with the model width. In this work, our primary objective is to verify the feasibility of TeGu. Thus, we adopt the standard SwiGLU FFN without aggressive compression. We fully agree that optimizing the structure of cMTPP and minimizing its parameter overhead is a valuable direction.
>
> ---
>
> We will incorporate the suggested revisions (including recommendations for formatting, wording, and presentation), along with the aforementioned supplementary details and experimental results, into the manuscript. If you have any other concerns, we are happy to engage in further discussion!

---

> > ### Author Rebuttal · Reviewer_9aAb · 2026-04-01
> >
> > Thanks for the additional experiments. However, my main concern remains unresolved, so I am keeping my original score. Specifically, regarding the first Soundness weakness, I do not think the results in Table 1 support the claims made in the results section. In my reading, TeGu does not meaningfully outperform its baselines except on IFEval. This is separate from my earlier point about statistical significance, though I appreciate the added experiments

---

> > > ### Author Response · Authors · 2026-04-04
> > >
> > > Thank you for your continued engagement. We respect your perspective on the magnitude of the absolute gains. However, we would like to take this opportunity to provide further context on how performance improvements are typically evaluated in the specific sub-field of inference-time decoding strategies, and to share brand-new results on the recently released Qwen3.5 models.
> > >
> > > ---
> > >
> > > **1. The Nature of Gains in Decoding Strategies**
> > > Unlike model fine-tuning or scaling up parameters, inference-time decoding strategies (like TeGu, DoLa, or standard Contrastive Decoding) do not inject new knowledge into the model. Instead, their goal is to better elicit existing capabilities. In this context, consistent improvements of 1% to 5% across diverse benchmarks, without catastrophic degradation on others, are generally considered highly meaningful and successful in the literature.
> > >
> > > **2. New Evidence: TeGu on Native MTP Models (Qwen3.5)**
> > > To further substantiate that TeGu provides consistent and meaningful improvements independently of our custom cMTPP training, we tested our method on the Qwen3.5 series released in early March 2026, which features native MTP heads out-of-the-box.
> > >
> > > Qwen3.5-0.8B-Base:
> > > Alpha|GSM8K|GSM8K-P|MATH500|Humaneval|Humaneval+|IFEval|MBPP
> > > -|-|-|-|-|-|-|-
> > > 0.0|46.55|39.29|3.00|21.95|**20.73**|25.88|25.40
> > > 0.1|**47.99**|39.70|4.00|20.12|18.90|27.36|24.60
> > > 0.2|47.31|**40.53**|3.80|21.34|20.12|28.28|25.00
> > > 0.3|47.92|39.29|**5.20**|**22.56**|**20.73**|**30.87**|**26.60**
> > >
> > > Qwen3.5-2B-Base:
> > > Alpha|GSM8K|GSM8K-P|MATH500|Humaneval|Humaneval+|IFEval|MBPP
> > > -|-|-|-|-|-|-|-
> > > 0.0|65.35|63.44|**19.20**|32.32|28.66|32.72|33.80
> > > 0.1|66.72|63.52|18.80|32.93|28.66|34.94|34.20
> > > 0.2|**66.87**|**65.26**|18.80|33.54|28.05|37.15|34.80
> > > 0.3|65.35|63.85|18.40|**34.76**|**29.27**|**37.34**|**35.60**
> > >
> > > Qwen3.5-4B-Base:
> > > Alpha|GSM8K|GSM8K-P|MATH500|Humaneval|Humaneval+|IFEval|MBPP
> > > -|-|-|-|-|-|-|-
> > > 0.0|83.55|83.95|11.80|49.39|42.68|37.89|51.60
> > > 0.1|84.46|84.37|13.40|52.44|45.12|42.51|52.00
> > > 0.2|**84.53**|82.88|14.00|51.83|45.73|47.32|52.40
> > > 0.3|84.15|**84.62**|**14.40**|**53.05**|**47.56**|**50.28**|**53.20**
> > >
> > > As shown above, purely as a zero-cost decoding intervention on state-of-the-art models, TeGu yields sustained improvements.
> > >
> > > While a +2% to +5% gain might seem numerically modest compared to scaling up a model by billions of parameters, achieving this for free at inference time by simply contrasting temporal contexts is precisely the contribution of this work. It offers a highly practical, plug-and-play methodology for the growing number of LLMs equipped with MTP heads.
> > >
> > > Finally, we respectfully emphasize that our primary contribution is a conceptual shift in the contrastive decoding paradigm, rather than solely chasing absolute state-of-the-art metrics. While MTP has traditionally been used primarily for speculative decoding (to improve speed), we uniquely repurpose it as a natural "amateur" signal to enhance generation quality. We believe this novel perspective, exploiting the temporal dimension for guidance, opens an elegant new direction for future decoding research.
> > >
> > > ---
> > >
> > > We sincerely thank you for pushing us to rigorously defend our work. While we may politely disagree on the threshold for "meaningful" absolute gains in the context of decoding algorithms, we deeply appreciate the time you have spent reviewing our paper. These new results and discussions will be included in the final manuscript to provide a comprehensive picture for the readers.

---

### Official Review · Reviewer_afic · 2026-03-12

**Soundness:** 3
**Presentation:** 3
**Significance:** 2
**Originality:** 4
**Overall Recommendation:** 5
**Confidence:** 4

**Summary:**

This paper proposes a novel single-model contrastive decoding method that has a multi-token predictor (MTP) as a premature distribution, typically used for speculative decoding in LLMs. Unlike middle-layer-based premature distribution, such as DoLa, the proposed method exploits the next-next token predictor from the previous step; prematureness is introduced by dropping the immediate previous token context. To obtain such an MTP for general LLMs without it, the authors propose a light-weight trainable conditioning layer attached before the output layer. Experiments across various benchmarks show that the proposed method with Qwen3 and Llama3 models consistently outperforms the baseline. They also demonstrate superior performance with an MTP-equipped open model, MiMo-7B, requiring no additional training. The experiments also consider the amateur's weight hyperparameter, the effect of knowledge distillation on MTP training, and differences in performance between factual retrieval and repetition-suppression behavior.

**Compliance With Llm Reviewing Policy:**

Affirmed.

**Final Justification:**

Thanks to the authors for the clarifications. I believe the paper is strong and I should be accepted. I keep my (already high) score.

**Key Questions For Authors:**

1. How did the authors set the DoLa's hyperparameters?
2. Why not KD-only training? Eq 5 suggests a mixing parameter $\lambda_{KD}$ but is not utilized in Section 5.1.
3. Why not use the trainable projection layer for early-exit from the mid-layer, similarly to DoLA? It would be better to conduct a controlled experiment that differs only in the source of amateur distribution.

**Limitations:**

yes

**Strengths And Weaknesses:**

## Soundness

Serving MTP as premature guidance in contrastive decoding is technically sound, supported by a mutual information-based formulation in the Appendix. The experimental results are sufficient to demonstrate the effectiveness of the proposed method. However, according to the experiments, existing contrastive decoding methods generally failed to outperform the non-contrastive baseline, and the authors did not explain why. It would be a more convincing comparison to search for optimal configurations and evaluation sets where the compared methods perform somewhat well; otherwise, analyze the reason why existing methods fail.

## Presentation

Very clear. The authors should address minor issues, primarily to improve reproducibility. Which MiMo-7B checkpoint among Base, SFT, and RL is used? How did the authors set the DoLa's hyperparameters? The vertical axis of Figure 2 looked arbitrarily modified for each benchmark. Normalize the values consistently, or show them in a table.

## Significance

Mixed results; depending on the tasks, significance is reduced. Now, even small, well-trained models already produce sufficiently contrastive distributions with a single head through extensive supervised and reinforcement learning. As the authors noted, the proposed method did not improve performance on factual-retrieval tasks, but perhaps better on long-horizon tasks. It would have been better if the analysis had been more in-depth.

## Originality

Utilization of MTP for contrastive decoding is a neat idea, clearly novel. The paper is the first attempt at contrastive decoding across different temporal contexts. Although the main experiments required additional training, the authors achieved good results with an off-the-shelf MTP model.

---

> ### Author Rebuttal · Authors · 2026-03-30
>
> We sincerely thank the reviewer for the highly positive evaluation, recognizing the novelty of our temporal contrastive decoding approach and the soundness of our theoretical formulation. Below is our point-by-point response to your questions and suggestions.
>
> ---
>
> > However, according to the experiments, existing contrastive decoding methods generally failed to outperform the non-contrastive baseline, and the authors did not explain why...
>
> We thank the reviewer for highlighting this. The underperformance of existing contrastive baselines primarily stems from scale mismatches and architectural bottlenecks:
> - Standard CD: The capacity gap between our selected amateur (Qwen3-0.6B) and the experts (1.7B/8B) may be suboptimal, leading to inappropriate penalization of the target distribution. (Nevertheless, Qwen3-0.6B is the smallest model in the Qwen3 series)
> - DoLa: In smaller models, representations across shallow and deep layers remain highly entangled. Furthermore, intermediate layer activations lack natural alignment with the LM head's vocabulary space.
>
> > Which MiMo-7B checkpoint among Base, SFT, and RL is used?
>
> Table 2 employs MiMo-7B SFT. We do not use the Base model because it exhibited abnormally low accuracy on certain benchmarks. Nonetheless, we also evaluate MiMo-7B RL, and the results indicate that TeGu can also yield performance improvements on this model.
>
> Alpha|Gsm8k|Gsm8k-P|HEval|HEval+|IFEval|Mbpp
> -|-|-|-|-|-|-
> 0.0|81.50|79.24|50.61|47.56|24.58|49.80
> 0.1|82.03|79.98|53.66|49.39|25.69|49.00
> 0.2|82.94|81.06|50.00|46.34|26.43|49.40
> 0.3|82.18|81.22|50.61|46.95|26.06|49.60
>
> Overall, TeGu demonstrates consistent performance improvements on both MiMo-7B SFT and MiMo-7B RL.
>
> > How did the authors set the DoLa's hyperparameters?
>
> We strictly adhered to the official recommended settings, utilizing `dola_layers="low"` as all our evaluated tasks involve long-form answers or reasoning. While the official recommendation for `repetition_penalty` is 1.2, we present results for both 1.0 and 1.2 in Table 1 to ensure a fair comparison.
>
> > Why not KD-only training? Eq 5 suggests a mixing parameter $\lambda_{KD}$ but is not utilized in Section 5.1.
>
> We use a hybrid objective because combining CE and KL losses accelerates convergence and yields slightly better final performance than KD-only training. The hard labels from the CE loss provide more explicit gradients.
>
> Regarding the mixing parameter, $\lambda_{KD} = 0.7$ was chosen as an effective empirical value. Our primary goal is to align the amateur distribution with the original model (hence the larger weight on KD), while utilizing CE to speed up training. This configuration already delivers highly satisfactory results without the need for exhaustive hyperparameter search.
>
> > Why not use the trainable projection layer for early-exit from the mid-layer, similarly to DoLA? ...
>
> We sincerely thank the reviewer for this insightful suggestion. We agree that using an identical projection architecture to isolate the amateur distribution source (spatial vs. temporal) rigorously validates our claims. We have added the following control experiment:
> We trained Early-Exit Projectors (EEP) on intermediate layers (3, 9, 15, 21, and 27) of Qwen3-1.7B. For a strictly fair comparison, EEP uses the exact same architecture as cMTPP (replacing time-step embedding with layer index) and the same CE + KD objective. We applied contrastive guidance with $\alpha=0.2$.
>
> Setting|GSM8K|GSM8K-P|Math500|HEval|HEval+|MBPP|IFEval
> -|-|-|-|-|-|-|-
> Baseline (Greedy)|72.48|70.64|12.60|40.24|37.20|42.60|15.16
> DoLa (Penalty 1.2)|72.93|71.05|9.20|28.05|25.00|24.80|26.06
> EEP (27)|72.70|70.38|12.20|38.41|36.59|41.80|16.82
> EEP (21)|73.23|70.80|12.60|39.63|36.59|43.20|19.22
> EEP (15)|74.75|71.13|12.40|39.63|37.20|44.60|21.07
> EEP (9)|74.52|70.88|12.20|40.24|37.80|44.40|19.59
> EEP (3)|74.14|72.29|12.20|38.41|36.59|44.20|19.96
> TeGu|75.51|72.21|12.80|42.07|38.41|45.20|26.99
>
> Main findings:
>
> - EEP validates our training design: EEP significantly mitigates DoLa's instability. It avoids the performance "collapse" seen in DoLa on coding tasks, proving the effectiveness of our conditional projection and hybrid loss.
> - Temporal > Spatial: Even under identical projection and training setups, TeGu (temporal) consistently outperforms the best EEP (spatial) across almost all benchmarks (e.g., 26.99 vs. 21.07 on IFEval, and 42.07 vs. 40.24 on HEval).
>
> Finally, TeGu offers superior system-level elegance. Unlike EEP or DoLa, which require intrusive intermediate-layer feature extraction that disrupts highly optimized inference frameworks (e.g., vLLM), TeGu simply reads the previous step's cache. For modern LLMs with native MTP heads, it is a nearly zero-cost enhancement.
>
> ---
>
> We will incorporate the suggested revisions (including Figure 2), along with the aforementioned supplementary details and experimental results, into the manuscript. If you have any other concerns, we are happy to engage in further discussion!

---

> > ### Author Rebuttal · Reviewer_afic · 2026-04-02
> >
> > Thanks to the authors for the clarifications.

---

> > > ### Author Response · Authors · 2026-04-04
> > >
> > > Thank you very much for your time, your insightful suggestions during the first round, and your confirmation that your concerns have been fully resolved. We are thrilled that you recognize the novelty of utilizing MTP for contrastive decoding and the elegance of our approach.
> > >
> > > ---
> > >
> > > As a quick update that aligns with your positive view on the future-proofing of our method: In early March 2026, the Qwen3.5 series was released, featuring native MTP heads across all model sizes. We immediately tested TeGu on Qwen3.5-0.8B, 2B, and 4B base models:
> > >
> > > Qwen3.5-0.8B-Base:
> > > Alpha|GSM8K|GSM8K-P|MATH500|Humaneval|Humaneval+|IFEval|MBPP
> > > -|-|-|-|-|-|-|-
> > > 0.0|46.55|39.29|3.00|21.95|**20.73**|25.88|25.40
> > > 0.1|**47.99**|39.70|4.00|20.12|18.90|27.36|24.60
> > > 0.2|47.31|**40.53**|3.80|21.34|20.12|28.28|25.00
> > > 0.3|47.92|39.29|**5.20**|**22.56**|**20.73**|**30.87**|**26.60**
> > >
> > > Qwen3.5-2B-Base:
> > > Alpha|GSM8K|GSM8K-P|MATH500|Humaneval|Humaneval+|IFEval|MBPP
> > > -|-|-|-|-|-|-|-
> > > 0.0|65.35|63.44|**19.20**|32.32|28.66|32.72|33.80
> > > 0.1|66.72|63.52|18.80|32.93|28.66|34.94|34.20
> > > 0.2|**66.87**|**65.26**|18.80|33.54|28.05|37.15|34.80
> > > 0.3|65.35|63.85|18.40|**34.76**|**29.27**|**37.34**|**35.60**
> > >
> > > Qwen3.5-4B-Base:
> > > Alpha|GSM8K|GSM8K-P|MATH500|Humaneval|Humaneval+|IFEval|MBPP
> > > -|-|-|-|-|-|-|-
> > > 0.0|83.55|83.95|11.80|49.39|42.68|37.89|51.60
> > > 0.1|84.46|84.37|13.40|52.44|45.12|42.51|52.00
> > > 0.2|**84.53**|82.88|14.00|51.83|45.73|47.32|52.40
> > > 0.3|84.15|**84.62**|**14.40**|**53.05**|**47.56**|**50.28**|**53.20**
> > >
> > > Without needing our cMTPP module, TeGu successfully provided consistent, zero-cost inference-time improvements across math, coding, and instruction-following tasks on these native MTP models (for instance, yielding nearly a 5% absolute improvement on IFEval for the 0.8B model, and consistent gains on Humaneval for the 4B model).
> > >
> > > ---
> > >
> > > Your initial feedback significantly helped us refine the presentation and empirical rigor of our work. We will ensure that the final  version incorporates all the control experiments (like the Early-Exit Projector comparison) and these new Qwen3.5 results.
> > > Thank you once again for your strong support of our work!

---

### Official Review · Reviewer_6dgD · 2026-03-23

**Soundness:** 3
**Presentation:** 2
**Significance:** 3
**Originality:** 3
**Overall Recommendation:** 4
**Confidence:** 4

**Summary:**

This paper focuses on contrastive decoding for large language models. Contrastive decoding compares the logits from a strong “expert” model and a weaker “amateur” model. It can enhance reasoning and factual signals while reducing common language priors. However, the authors point out that previous methods have problems such as high computational cost and instability. To solve this, they propose Temporal Guidance (TeGu). Instead of using two different models, TeGu uses the same model with different context lengths to construct the expert and amateur distributions. Experiments show that TeGu outperforms greedy decoding, standard CD, and DoLa on tasks like math, coding, and instruction following.

**Compliance With Llm Reviewing Policy:**

Affirmed.

**Final Justification:**

After reading the other reviewers’ comments and the authors’ rebuttal, I maintain my weak accept recommendation.

The authors discuss the theme of improving contrastive decoding through temporal guidance, and overall, the authors outline a meaningful problem in designing efficient self-contrast mechanisms for LLMs. The motivation is clear, and the method is reasonable, with some insightful and inspiring ideas. The experiments show generally positive results across several benchmarks.

There is still room to improve how well the method works across different types of tasks and models, but I consider this a qualified piece of work.

Therefore, I am inclined to keep my original weak accept decision.

**Key Questions For Authors:**

Refer to Cons

**Limitations:**

yes

**Strengths And Weaknesses:**

Pros:
1. The problem is well-defined, and the motivation is clear. The proposed method is well aligned with the motivation.
2. The issues mentioned in the motivation (such as efficiency) are supported by experiments. The results clearly show the effectiveness of the method.
3. The idea that LLMs exhibit local preference is interesting. Temporal Guidance is a novel idea and is related to some classical ideas in computer systems.

Cons:
1. Figure 1 is not clear. The lines are messy and hard to read.
2. Many LLMs do not support native MTP. In such cases, the performance depends heavily on the accuracy of cMTPP. However, the paper does not provide direct evaluation of cMTPP quality.
3. The paper treats predictions with less context as the amateur. However, this does not always mean lower quality. Some tokens do not strongly depend on local context. In such cases, the method may not work well. This assumption works in many tasks, but it is not universal.
4. The authors claim that DoLa is unstable on small models. However, their method also shows sensitivity to hyperparameters on small models, which may affect robustness.

---

> ### Author Rebuttal · Authors · 2026-03-30
>
> We sincerely thank the reviewer for recognizing the clarity of our motivation, the effectiveness of our approach, and the novelty of connecting local preference with temporal guidance. We address your valuable feedback point-by-point below.
>
> ---
>
> > Many LLMs do not support native MTP. In such cases, the performance depends heavily on the accuracy of cMTPP. However, the paper does not provide direct evaluation of cMTPP quality.
>
> We sincerely thank the reviewers for this insightful comment. We agree that providing a direct evaluation of cMTPP would be valuable for elucidating its internal behavior. We evaluated the representations generated by cMTPP with varying degrees of offset on the Wikitext-2 test set, using both the Qwen3-1.7B and Qwen3-8B models.
>
> Model|Module / Offset|Top-1 Acc|Top-5 Acc|Cross-Entropy|PPL
> -|-|-|-|-|-
> Qwen3-1.7B|Base NTP (Expert)|48.95%|71.96%|2.8136|16.67
> ||cMTPP Offset-2|16.23%|33.31%|6.2004|492.94
> ||cMTPP Offset-3|8.63%|21.70%|7.2803|1451.44
> ||cMTPP Offset-4|5.78%|17.18%|7.6812|2167.31
> Qwen3-8B|Base NTP (Expert)|53.96%|77.16%|2.2737|9.72
> ||cMTPP Offset-2|20.37%|39.19%|5.5990|270.16
> ||cMTPP Offset-3|10.56%|25.10%|6.8575|950.95
> ||cMTPP Offset-4|6.88%|19.31%|7.3668|1582.53
>
> - Expected Performance Degradation: As the offset increases, the Top-1 accuracy decreases and the perplexity rises. This confirms that cMTPP successfully constructs an amateur distribution.
> - Capacity Scaling: Compared to Qwen3-1.7B, Qwen3-8B achieves higher accuracy across all offsets. This indicates that cMTPP effectively leverages the stronger representation capacity of the larger base model while maintaining the relative performance gap necessary for contrastive decoding.
>
> This demonstrates that cMTPP generates a robust, performance-degrading probability distribution, making the TeGu method effective. We will include this quantitative evaluation and the associated discussion in the supplementary material of the final version.
>
> > The paper treats predictions with less context as the amateur. However, this does not always mean lower quality. Some tokens do not strongly depend on local context. In such cases, the method may not work well. This assumption works in many tasks, but it is not universal.
>
> We completely agree with your insightful observation. Indeed, contrastive decoding methods generally face the challenge that an "amateur" model may not uniformly perform worse than the "expert" across all tokens.
>
> To mitigate this in our current implementation, we employ the Adaptive Plausibility Constraint (APC) (mentioned in Section 3.3) as a safeguard, ensuring that TeGu does not improperly penalize plausible tokens by truncating the distribution tail. Furthermore, your comment highlights a highly promising direction for future research. Since the dependency on local context varies significantly across different tasks and token types, future work could design adaptive, task-specific mechanisms to dynamically modulate how the context-deficient MTP predictions are utilized for guidance, rather than using a static contrastive strategy.
>
> We will incorporate this valuable discussion regarding the non-universal nature of the local-context dependency assumption, along with these potential future directions, into our Limitations section.
>
> > The authors claim that DoLa is unstable on small models. However, their method also shows sensitivity to hyperparameters on small models, which may affect robustness.
>
> We aim to clarify the distinction between DoLa's "instability" and TeGu's "sensitivity."
>
> When we refer to DoLa's instability on small models, we are addressing the severe "decoding collapse" that it exhibits. For instance, on the Qwen3-1.7B model, DoLa's performance on the MBPP benchmark plummets to 24.80%, a significant drop from the 42.60% achieved by the greedy decoding baseline.
>
> In contrast, while TeGu also exhibits sensitivity to the hyperparameter $\alpha$ on 1.7B models, its performance degrades gradually and smoothly as $\alpha$ increases. As illustrated in Figure 4, even with suboptimal settings, TeGu consistently outperforms the baseline and does not manifest a collapse phenomenon analogous to that observed in DoLa.
>
> > Figure 1 is not clear. The lines are messy and hard to read.
>
> We sincerely appreciate the reviewer's feedback regarding Figure 1. In the revised version, we will optimize this figure to ensure that the presentation of TeGu is intuitive, clear, and easy to read.
>
> ---
>
> We will incorporate the suggested revisions, along with the aforementioned supplementary details and experimental results, into the manuscript. If you have any other concerns, we are happy to engage in further discussion!

---

> > ### Author Rebuttal · Reviewer_6dgD · 2026-04-03
> >
> > Thanks to the authors for their responses, but the experimental design of the cMTPP assessment is not sufficiently convincing, so I will maintain my original score.

---

> > > ### Author Response · Authors · 2026-04-04
> > >
> > > Thank you for your prompt response and for sharing your remaining concern regarding the cMTPP assessment. We completely understand your perspective: evaluating a custom-trained projection module introduces confounding variables that might obscure the fundamental effectiveness of the TeGu method itself.
> > >
> > > To provide the most direct and convincing evidence that our core hypothesis (Temporal Guidance) is robust and independent of cMTPP's design, we have leveraged a timely advancement in the open-source community. In early March 2026, the Qwen3.5 model series was officially released, featuring **native MTP heads** across all model scales out-of-the-box.
> > >
> > > This allowed us to test TeGu purely as an inference-time decoding strategy on native MTP architectures, completely bypassing cMTPP. We evaluated TeGu on Qwen3.5-0.8B, 2B, and 4B base models:
> > >
> > > **Performance on Native MTP Models (Qwen3.5 Series):**
> > >
> > > Qwen3.5-0.8B-Base:
> > > Alpha|GSM8K|GSM8K-P|MATH500|Humaneval|Humaneval+|IFEval|MBPP
> > > -|-|-|-|-|-|-|-
> > > 0.0|46.55|39.29|3.00|21.95|**20.73**|25.88|25.40
> > > 0.1|**47.99**|39.70|4.00|20.12|18.90|27.36|24.60
> > > 0.2|47.31|**40.53**|3.80|21.34|20.12|28.28|25.00
> > > 0.3|47.92|39.29|**5.20**|**22.56**|**20.73**|**30.87**|**26.60**
> > >
> > > Qwen3.5-2B-Base:
> > > Alpha|GSM8K|GSM8K-P|MATH500|Humaneval|Humaneval+|IFEval|MBPP
> > > -|-|-|-|-|-|-|-
> > > 0.0|65.35|63.44|**19.20**|32.32|28.66|32.72|33.80
> > > 0.1|66.72|63.52|18.80|32.93|28.66|34.94|34.20
> > > 0.2|**66.87**|**65.26**|18.80|33.54|28.05|37.15|34.80
> > > 0.3|65.35|63.85|18.40|**34.76**|**29.27**|**37.34**|**35.60**
> > >
> > > Qwen3.5-4B-Base:
> > > Alpha|GSM8K|GSM8K-P|MATH500|Humaneval|Humaneval+|IFEval|MBPP
> > > -|-|-|-|-|-|-|-
> > > 0.0|83.55|83.95|11.80|49.39|42.68|37.89|51.60
> > > 0.1|84.46|84.37|13.40|52.44|45.12|42.51|52.00
> > > 0.2|**84.53**|82.88|14.00|51.83|45.73|47.32|52.40
> > > 0.3|84.15|**84.62**|**14.40**|**53.05**|**47.56**|**50.28**|**53.20**
> > >
> > > Key Takeaways:
> > >
> > > 1. **Consistent Gains Without cMTPP:** On models with natively trained MTP heads, TeGu consistently improves performance across reasoning, coding, and instruction-following benchmarks, validating our core theoretical claim.
> > >
> > > 2. **Robustness Across Scales:** TeGu shows stable improvements even on the highly compact Qwen3.5-0.8B (e.g., IFEval jumps from 25.88% to 30.87%).
> > > We believe these new results on state-of-the-art native MTP models directly address your concern by proving that TeGu is a fundamentally highly effective decoding strategy, and its success is not an artifact of cMTPP. With the rapid adoption of native MTP in modern LLMs (like Qwen3.5), TeGu can be deployed immediately as a zero-cost enhancement.
> > >
> > > We would also like to clarify that TeGu's core contribution is leveraging the temporal lag of MTP for decoding guidance, rather than the specific cMTPP architecture. cMTPP was merely an engineering compromise because open-source native MTP models were scarce at the time of writing. TeGu's ideal use case is natively MTP-equipped models, and its consistent success on Qwen3.5 effectively validates our core hypothesis independently of cMTPP.
> > >
> > > ---
> > >
> > > We hope this new, strong empirical evidence alleviates your remaining concerns. We will prominently feature these Qwen3.5 results in the final version of the paper. Thank you again for pushing us to strengthen our empirical validation!

---

### Decision · Program_Chairs · 2026-04-30

**Decision:**

Reject

**Comment:**

This paper proposes a single-model contrastive decoding framework that leverages temporal discrepancies via multi-token prediction (MTP) to construct an internal “amateur” distribution, aiming to eliminate auxiliary models and reduce computational overhead. While the idea of exploiting temporal misalignment as a contrastive signal is interesting and conceptually appealing, the current work falls short of providing sufficiently convincing evidence for its effectiveness and robustness.

The empirical evaluation, although showing some improvements across tasks and models, lacks clarity in key experimental details, and important design choices are insufficiently justified. In particular, concerns raised about the interpretability and presentation of Table 1 and the overall experimental setup are not adequately addressed in the rebuttal, leaving ambiguity about whether the observed gains truly stem from the proposed mechanism or from confounding factors. Although the rebuttal attempts to strengthen the claims with additional experiments and statistical tests, these additions are incremental and do not fully resolve the core issues regarding evaluation rigor and clarity. Given these limitations, the paper does not yet meet the bar for acceptance in its current form.